# Etiology of Metabolic Syndrome and Dietary Intervention

**DOI:** 10.3390/ijms20010128

**Published:** 2018-12-31

**Authors:** Hang Xu, Xiaopeng Li, Hannah Adams, Karen Kubena, Shaodong Guo

**Affiliations:** 1Department of Nutrition and Food Science, College of Agriculture and Life Sciences, Texas A&M University, College Station, TX 77843, USA; hangxu@tamu.edu (H.X.); lixiaopeng@tamu.edu (X.L.); hannahbadams2018@tamu.edu (H.A.); k-kubena@tamu.edu (K.K.); 2College of Food Science and Technology, Huazhong Agricultural University, Wuhan 430070, China

**Keywords:** metabolic syndrome, obesity, insulin resistance, personal nutrition, dietary intervention

## Abstract

The growing prevalence of metabolic syndrome (MetS) in the U.S. and even worldwide is becoming a serious health problem and economic burden. MetS has become a crucial risk factor for the development of type 2 diabetes mellitus (T2D) and cardiovascular diseases (CVD). The rising rates of CVD and diabetes, which are the two leading causes of death, simultaneously exist. To prevent the progression of MetS to diabetes and CVD, we have to understand how MetS occurs and how it progresses. Too many causative factors interact with each other, making the investigation and treatment of metabolic syndrome a very complex issue. Recently, a number of studies were conducted to investigate mechanisms and interventions of MetS, from different aspects. In this review, the proposed and demonstrated mechanisms of MetS pathogenesis are discussed and summarized. More importantly, different interventions are discussed, so that health practitioners can have a better understanding of the most recent research progress and have available references for their daily practice.

## 1. Introduction

Metabolic syndrome (MetS), is a complex of metabolic abnormalities, which serves as a risk factor for type 2 diabetes mellitus (T2D) and cardiovascular diseases (CVD). The main characteristic components include hyperglycemia, raised blood pressure, elevated triglyceride levels, low HDL-cholesterol levels, and obesity (particularly central adiposity). Since 1988, when Reaven first described it as “Syndrome X” [1], the definition and diagnostic criteria have been proposed and modified several times by different public health organizations. This definition will continue to develop as our ability to predict the metabolic consequences, in regard to diabetes and cardiovascular disease also develops [2]. In the development of the definitions, the debate has been focused on whether obesity or insulin resistance is the unifying feature and underlying cause for MetS. Indeed, MetS has been renamed “Insulin Resistance Syndrome” by the European Group for Study of Insulin Resistance (EGIR) in 1999 and American Association of Clinical Endocrinologists (AACE) in 2003. Obesity and insulin resistance are further discussed in the next section of this review.

Recently, the criteria published by American Heart Association/National Heart, Lung and Blood Institute (AHA/NHLBI) (slightly revised from National Cholesterol Education Program, Adult Treatment Panel III NCEP-ATP III) and International Diabetes Federation (IDF) have been widely used in the U.S. and worldwide [2]. The historical definitions and criteria for diagnosis are summarized and organized in Table 1. In 2005, IDF dropped the World Health Organization (WHO) requirement for insulin resistance, and requires obesity be prerequisite to diagnose MetS, whereas AHA did not mandate abdominal obesity as a required risk factor [3]. Dr. Reaven has criticized IDF for its emphasis on obesity rather than insulin resistance, and considers insulin resistance as more likely to contribute to MetS [4]. In 2009, IDF and AHA/NHLBI representatives held discussions and finally agreed on a definition of MetS. Abdominal obesity would not be an obligatory component, but waist circumference would continue to be a useful screening tool. The presence of any three out of five abnormal findings diagnoses a person with MetS. As for the waist circumference thresholds for abdominal obesity, different organizations still hold relatively different standards [3], as shown in Table 1. Regardless, the risk associated with a waist circumference will differ in different populations with gender and ethnicity. Further studies are needed/encouraged to explore the relation of waist circumference thresholds to metabolic risk and cardiovascular outcomes in different populations [3].

The prevalence of MetS keeps growing with the rising rates of obesity worldwide, no matter what diagnostic criteria is being used. According to the Centers for Disease Control and Prevention (CDC), from 2011 to 2014, over one-third of adults and around 17% in youth in the United States were obese [5]. Based on WHO data, worldwide obesity has nearly tripled since 1975 [6]. In line with obesity trends, around 34% of adults have MetS [7,8]. Metabolic syndrome is a crucial contributor to type 2 diabetes and cardiovascular disease. In co-occurrence, the prevalence of CVD and T2D is also increasing. Heart disease and diabetes still take the 1st and 7th place of the leading causes of death in the U.S. based on CDC data for 2015 [9]. Thus, it is important to undercover the mechanisms, and develop effective intervention strategies accordingly, so as to control the prevalence of MetS and prevent development into diabetes and CVD.

## 2. Etiology of MetS

Although research has been carried out in recent decades on MetS, the exact underlying etiology is still not completely understood. Many contributing factors and mechanisms have been proposed, including insulin resistance, adipose tissue dysfunction, chronic inflammation, oxidative stress, circadian disruption, microbiota, genetic factors, and maternal programming, etc. The major contributors are discussed in the following paragraphs.

### 2.1. Insulin Resistance

MetS is also widely known as insulin resistance syndrome [10,11] due to the causative role insulin resistance plays in the syndrome [12,13]. Even the group European Group for the Study of Insulin Resistance (EGIR) and American Association of Clinical Endocrinologists (AACE) used the term insulin resistance syndrome rather than metabolic syndrome. Because insulin resistance is difficult to evaluate directly, especially in a clinical setting, some types of evidence were accepted, including impaired fasting glucose (IFG), impaired glucose tolerance (IGT) and type 2 diabetes mellitus. Some other factors were also served as diagnostic criteria, such as blood pressure, obesity, and disordered lipid profile (Table 1).

In order to understand insulin resistance, it is important to understand the basis of insulin action and its signaling cascades. In a normal situation, an increase in blood glucose upon feeding stimulates insulin release from pancreatic β cells. Insulin, together with glucose, stimulates glucose uptake from circulation into cells for glycolysis or is stored as glycogen in the liver, muscle, or adipose. This results in the suppression of hepatic gluconeogenesis. All these physiological processes work together to bring down the blood glucose to the normal basal level range. GLUT4 is one of the most important glucose transporters, mainly expressed in muscle and adipose tissue. Under the stimulation of insulin, GLUT4 is mobilized from the cytosol to the cell membrane to transport glucose from outside of the cell to inside. This is the rate-limiting step in glucose uptake and muscle glycogen synthesis [14,15]. Insulin not only regulates glucose metabolism, but also modulates lipid metabolism. Lipogenesis is enhanced in response to insulin, while lipolysis is inhibited.

In an abnormal, or insulin-resistant state, there is a loss of initial insulin secretion (first phase) in response to a glucose load, resulting in postprandial hyperglycemia. Subsequently, an exaggerated second-phase insulin response causes chronic hyperinsulinemia. Insulin-responsive tissues cannot sensitize or respond to insulin efficiently. Insulin-mediated glucose uptake, glycolysis, and glycogen synthesis are all impaired. Over time, insulin resistance worsens and pancreatic β-cells gradually become stressed, fatigued and apoptotic, until they completely lose their function [16,17]. Without insulin, hyperglycemia persists. Thus, the late-stage type 2 diabetic patients are similar to type 1 diabetic patients, who solely rely on external insulin injection to bring down blood glucose to a relatively normal level. As the disease progresses, even insulin injection may not help with blood glucose control, due to severe insulin resistance in the late stage. Other intervention strategies need to be developed to improve insulin sensitivity or β-cell function, such as diet and physical activity.

### 2.2. Pancreatic β-Cell Dysfunction

The β-cell function (BCF) is closely related with MetS. A study of the Cleveland clinic suggested that β-cell dysfunction (estimated using the disposition index—DI) is highly correlated with the severity of MetS (estimated using z-score) independent of sex, body fat, blood lipids, blood pressure, insulin resistance, and glucose metabolism [18]. Therefore, improving BCF can be an important strategy to ameliorate MetS. It is suggested that increased cardiorespiratory fitness (CRF) is positively associated with enhanced BCF in individuals with MetS, independent of body fat%, and other confounding factors [19]. The researchers recommended that “Equal, if not more attention should be dedicated to CRF improvement relative to fat-loss for favorable pancreatic BCF and thus a possible reduction in CVD risk in individuals with MetS” [19].

In recent years, as weight-loss strategies have been evaluated, CRF has attracted more attention. A meta-analysis of fitness (CRF) and fatness (BMI) on overall mortality showed that unfit individuals (low CRF) had twice the risk of mortality compared to normal weight-fit individuals, regardless of BMI. It is suggested that fitness-based interventions rather than weight-loss driven strategies reduce mortality risk overall [20]. It has been repeatedly shown that CRF is a powerful prognostic factor in all populations of coronary heart disease (CHD) and cardiovascular (CV) patients [21]. Higher levels of CRF can be protective, and its improvement may yield better benefits [22]. Lean body mass is a better indicator for longevity or mortality than BMI, therefore more attention needs to be given to lean mass or CRF rather than body weight as a strategy for lowering the risk of MetS [23,24,25].

### 2.3. Cellular Dysfunction by Protein Kinases and Phosphatases

Normally, insulin can bind to the insulin receptor in the cell membrane, resulting in excitation of insulin receptor tyrosine kinase. Subsequently, insulin receptor substrate-1 (IRS1) and -2 (IRS2), are recruited and phosphorylated on the tyrosine sites to continue to phosphorylate the downstream target signaling proteins, either phosphatidylinositide 3-kinases (PI3K) or a class of small GTPase (RAS), which are two major pathways in insulin-mediated activities. PI3K→Akt pathway is the major channel of the metabolic effects of insulin. Phosphorylated PI3K catalyzes the generation of phosphatidylinositol 3,4,5-trisphosphate (PIP3) from phosphatidylinositol 4,5-trisphosphate (PIP2), resulting in phosphoinositide-dependent protein kinase (PDK1/PDK2) and Akt binding to PIP3 [26,27]. PDK1/PDK2 then phosphorylates Akt [2], which phosphorylates a number of downstream targets to mediate the effect of insulin on enhancing GLUT4 translocation, glycogen synthesis, protein synthesis, and lipogenesis, as well as inhibiting apoptosis and hepatic gluconeogenesis. Some of these metabolic effects work through Akt phosphorylation of FOXO1. FOXO1 is required in the nucleus for the transcription of some of the gluconeogenic and lipogenic genes. Upon phosphorylation by Akt, FOXO1 translocates from nucleus to cytosol, suppressing glucose production in the liver and promoting cell survival in the heart [28]. Many of these phosphorylation events are used as indicators of insulin sensitivity [13]. RAS→mitogen-activated protein kinase (MAPK) signaling pathway mainly mediates the effect of insulin on mitogenesis and cell growth (Figure 1).

Under insulin resistance, the phosphorylation signaling pathway becomes impaired, which leads to decreased GLUT4 expression, or dysfunction of translocation, resulting in impaired glucose transport, suppressed glycogen storage, and inhibited protein synthesis. GLUT4 activation by insulin is also important during the glucose disposal, and p38 MAPK may be involved in this process [29]. Under insulin resistance state, both GLUT4 translocation and activation are affected [30]. Meanwhile, deficits in insulin signaling pathway release FOXO1 back to the nucleus to promote the expression of gluconeogenic genes and a rise in very-low-density lipoprotein (VLDL) secretion [15,31]. The real mechanism of insulin resistance is still not completely understood. However, many factors have been shown to interact with each other, and contribute to insulin resistance. For example, hyperinsulinemia results in serine/threonine phosphorylation of IRS (which promotes IRS degradation) and prevention of tyrosine phosphorylation (which is the classic phosphorylation in insulin signaling pathways involved in PI3K→Akt→FOXO1). Proinflammatory cytokines such as tumor necrosis factor α(TNFα) and genetic defects (i.e., Akt) induce insulin resistance.

Insulin resistance locally in the key insulin-responsive tissues, such as adipose tissue, liver, muscle, brain, immune cells and intestine cells, works alone or synergistically towards systemic insulin resistance. Guo has reviewed the mechanism of insulin resistance in different tissues [13]. He has found that central nervous system (CNS) insulin resistance is the main cause of obesity by regulating appetite and food intake behavior; insulin resistance in adipose tissue results in hyperlipidemia and inflammation; hepatic insulin resistance causes hyperglycemia; cardiac insulin resistance promotes heart failure; pancreatic insulin resistance results in impaired β-cell regeneration; insulin resistance in vascular endothelium promotes hypertension and disrupts glucose homeostasis; insulin resistance in skeletal muscle shortens lifespan, and insulin resistance in bone impairs glucose homeostasis [13].

Insulin signaling is also governed by phosphatases. Phosphorylated tyrosine residues in IRS1 and 2 can be dephosphorylated by protein tyrosine phosphatase 1B (PTP1B) and T cell protein tyrosine phosphatase (TCPTP) which results in termination of insulin signaling. Both of these have been proposed to be potential therapeutic targets due to their inhibitory effect on insulin signaling [32]. In the pancreas, FOXO1 promotes beta-cell differentiation and insulin secretion [33], probably contributing to hyperinsulinemia in T2D. Phosphorylated FOXO1-S253 can be dephosphorylated by protein phosphatase 2A (PP2A), MAPK phosphatase-3 (MKP3), or a nuclear phosphatase SCP4. Thus, suppressing the activities of these protein phosphatases may enhance FOXO1-S253 phosphorylation, suppression of FOXO1, and hepatic glucose production [34].

### 2.4. Suppression of IRS1 and IRS2 Gene Expression and Function

IRS1 and IRS2 have crucial roles in the insulin signaling cascade. Systemic deletion of both IRS1 and IRS2 causes embryonic lethality in mice [35]. Dysfunction of IRS1 and IRS2 in different tissues contributes local, or even systemic insulin resistance, and pathogenesis of metabolic diseases [13]. For example, loss of IRS1 and IRS2 in the heart causes impaired insulin signaling and heart failure. It has been suggested that chronic hyperinsulinemia activates p38 (p38α MAPK mitogen-activated protein kinase) which can reduce IRS1 and IRS2 proteins by promoting their ubiquitination and/or degradation, resulting in insulin resistance [36]. During insulin resistance in rodents and humans, glucose uptake mediated by IRS1 was severely impaired whereas salt reabsorption in kidney proximal tubule mediated by IRS2 was reserved. This explains how insulin resistance results in a state of salt overload, leading to hypertension [37]. Research has found that APPL1 (adaptor protein containing pleckstrin homology domain, phosphotyrosine binding (PTB) domain, and leucine zipper motif) serves as a binding partner of IR and IRS proteins [38]. Its phosphorylation at Ser401, which is downregulated in obesity, plays a key role in regulating the interaction of IR and IRS proteins, and thus insulin signaling [38]. Adiponectin stimulates APPL1 Ser401 phosphorylation to promote insulin signaling. Adiponectin, however, also stimulates skeletal muscle autophagy and antioxidant potential to reduce insulin resistance during high fat-diet (HFD) feeding in mice [39].

Factors that interfere with either expression or phosphorylation of IRS1 and IRS2 may contribute to insulin resistance. Direct evidence has shown that mammalian PTEN (phosphatase and tensin homolog) is a dual-specificity protein phosphatase, dephosphorylating tyrosine-, serine- and threonine-phosphorylated proteins. Also, it acts as a lipid phosphatase, removing the phosphate in the D3 position of the inositol ring from PIP3. Similar to PTP1b, PTEN is also a protein tyrosine phosphatase for IRS1, and the dephosphorylation of IRS1 results in impaired insulin signaling. On the other hand, NEDD4 (neural precursor cell-expressed developmentally downregulated protein 4) ubiquitin ligase can work on PTEN resulting in its ubiquitination and degradation. Thus, as the antagonist of PTEN, NEDD4 promotes IRS1 phosphorylation and therefore insulin signaling [40]. It has been suggested that long-term high dose statins (e.g., rosuvastatin) can induce insulin resistance by upregulating PTEN in skeletal muscle [41]. The high-molecular-mass complexes containing insulin receptor substrates also involved in mediating and regulating insulin-like activities were summarized in a previous review [42]. Not only insulin-like growth factors (IGF)/insulin but also other cytokines/hormones contribute to the formation of IRSs associated with other proteins (IRSAPs). IRSAPs can regulate IGF/insulin signaling pathway by controlling IRS tyrosine phosphorylation and interaction with PI3K [43,44,45]. IRSAPs also play important roles in the modification of IRSs stability, intracellular localization, and RNA metabolism and translation [42].

Both systemic IRS1 null mice and IRS2 null mice displayed insulin resistance, indicating both are irreplaceable [46,47,48]. When IRS1 is phosphorylated on serine/threonine sites, IRS1 both tyrosine phosphorylation and downstream insulin signaling would be hindered. High fat-diet (HFD) produces insulin resistance in the hippocampus of mice by increasing serine-phosphorylated IRS1 (IRS1-pS616), resulting in insulin resistance [49]. Hepatitis C virus NS5A promotes insulin resistance and increases gluconeogenesis through IRS1 serine phosphorylation (Ser307) followed by decreased phosphorylation of Akt-Thr308, FOXO1-Ser256, and GSK3β-Ser9—all downstream players of the insulin signaling pathway [50].

Overexpression of IRS1 in endothelial cells restored angioblast differentiation and wound healing in HF-induced diabetic mice with insulin resistance. Hence, endothelial IRS1 can serve as a potential target to improve angiogenesis, and wound healing in patients with diabetes and obesity [51]. MEMO1 (mediator of ErbB2-driven cell motility 1), a new IRS1-interacting protein, was discovered to bind IRS1 and activate the downstream PI3K and Akt signaling pathway, leading to epithelial-mesenchymal transition in mammary epithelial cells. Therefore, MEMO1 acts as an oncogene, and is a potential therapeutic target for cancer treatment [43]. Meta-analysis of several human studies indicated that IRS1 variants rs7578326 G-allele carriers and rs2943641 T-allele carriers had a lower risk of insulin resistance, T2D, and MetS [52].

Several lines of new evidence showed that IRS1 is also a target of microRNAs. MiR-128a regulates myogenesis by targeting IRS1/Akt insulin signaling [53]. MiR-145 also down-regulates IRS1 expression and its downstream Akt/FOXO1 signaling, which suppresses hepatocellular carcinoma [54]. MiR-126 directly interacts with IRS1 to mediate the repression of IRS1 translation [55]. IRS1 downregulation can be programmed in offspring of obese mice. A research group found that maternal diet-induced obesity leads to offspring having increased levels of MiR-126 which targets IRS1 and adipose tissue insulin resistance prior to the development of obesity, resulting in increased risk of T2D [55].

IRS2 is especially crucial in BCF and the hypothalamus. The β cell- and hypothalamus-specific knockout of IRS2 in mice induced obesity and both leptin and insulin resistance [56]. Endothelial cell-specific IRS2 knockout mice exhibited decreased pancreatic islet blood flow, causing impaired glucose-induced insulin secretion. Thus, IRS2 in endothelial cells may serve as a novel therapeutic target for restoring β-cell function, and ameliorating glucose intolerance in MetS [57].

Serine/threonine phosphorylation of IRS2 impairs normal IRS2 tyrosine phosphorylation involved in insulin signaling. Angiotensin II and protein kinase C can phosphorylate IRS2 on Ser303 and Ser675 sites to inhibit insulin-induced IRS2-Tyr911 phosphorylation in endothelial cells, hindering its anti-atherogenic actions (pAkt/endothelial nitric oxide) [58]. It has been demonstrated that IRS2-Ser1137/1138 are novel cAMP-dependent phosphorylation sites, which allows IRS2 to bind to 14-3-3 proteins for degradation [59]. Important in the insulin signaling pathway, IRS2 was identified as a likely driver oncogene which activates the oncogenic PI3 kinase pathway and increases cell adhesion, both characteristics of invasive colorectal cancer cells [60].

### 2.5. Obesity and Lipid Toxicity

Obesity is closely related to a variety of chronic diseases, such as CVD, T2D, NAFLD, and cancer. Overnutrition and physical inactivity together contribute to energy imbalance, in which energy intakes overpass energy expenditures, resulting in fat storage in obese individuals. High, non-esterified fatty acids (NEFA) are almost always observed. These have been shown to be an important contributor to insulin resistance and inflammation. Saturated fatty acids, such as palmitate, reduces IRS1, 2 tyrosine phosphorylation, promotes FOXO1 activity, and induces serine/threonine phosphorylation by activation of intracellular protein kinases, such as protein kinase C (PKC) and c-Jun N-terminal kinase (JNK) [61].

Adipose can secrete some adipokines that can communicate with other different tissues, including the brain, immune cells etc. For example, leptin is secreted by adipocytes, and signals the brain for satiety. Normally, as fat cells expand, more leptin is secreted to the brain to signal the termination of eating behavior. However, leptin resistance can happen in obese individuals, similar to insulin resistance. In these individuals, even high levels of leptin cannot create satiety [62]. Leptin may also regulate glucose homeostasis, pancreases β cells, and insulin-sensitive tissues [63,64].

With the growth of adipose tissue during obesity development, angiotensinogen (Agt), a precursor of angiotensin II that enhances the sympathetic nervous system and blood pressure, is drastically overexpressed [65,66]. Agt is suggested to be a FOXO1-target gene in the liver [67]. Thus, failure of insulin-suppressed FOXO1 may promote AngII production that increases SNS and blood pressure.

### 2.6. Oxidative Stress and Glucose Toxicity

Oxidative stress, defined as an imbalance in the production and degradation of ROS, is closely associated with MetS, leading to carcinogenesis, obesity, diabetes, and CVD [68]. Increased low density lipoprotein (LDL), and decreased high density lipoprotein (HDL) levels are frequently observed in an environment of oxidative stress. Respiratory circuit occurs in the Mitochondria and uses reducing equivalents generated from the tricarboxylic acid cycle (TCA) cycle and oxygen to produce adenosine triphosphate (ATP), and water through the electron transport chain (ETC). It is estimated that up to 2% oxygen consumed can be diverted to the production of reactive oxygen species (ROS) formation by mitochondria [69], which can be utilized and balanced out by the anti-oxidative system in a normal state. A high-energy diet could increase the metabolic load of the mitochondria resulting in an overactive ETC that can form excessive ROS as the by-product. ROS contributes to mitochondrial damage affecting normal cellular signaling and metabolic processes. TNFα and free fatty acids (FFA) can also be linked with oxidative stress and inflammation. Obesity in animal models, both diet-induced and genetic, has shown overexpression of NOX (NADPH oxidase) subunits which positively correlates with increased oxidative stress in MetS. Some evidence has shown that abnormal generation of ROS can induce adipogenesis via pre-adipocytes proliferation and differentiation and therefore contributes to the development of obesity and MetS. ROS serves not only as the trigger, but also the outcome of obesity. Available evidence shows that obesity can cause systemic oxidative stress through NOX activation, endoplasmic reticulum (ER) stress in adipocytes, and excessive ROS production subsequent to high-fat high-carbohydrate diet and suppressed anti-oxidative system [70,71]. Oxidative stress in metabolic disorder leads to diabetes and CVD. The elevated levels of glucose can cause mitochondrial dysfunction, such as an increase in ROS production and insulin resistance. ROS also induces beta-cell dysfunction, defective proliferation, and growth.

Glucose flux through the hexosamine biosynthetic pathway (HBP) causes the post-translational modification of cytoplasmic and nuclear proteins by O-linked beta-*N*-acetylglucosamine (*O*-GlcNAc), which serves as a nutrient sensor for control of insulin signaling in cells. For example, glucose and OGT-mediated glycosylation of Akt at Thr-308 can prevent the Akt-Thr308 phosphorylation by insulin signaling. This also provides a mechanism by which hyperglycemia can induce insulin resistance at the molecular level [72]. Therefore, therapeutic strategies to overcome glucose toxicity and stress-induced metabolic abnormalities can be feasible by control of HBP. Exercise can also improve the antioxidant system of the body, which helps manage the oxidative stress by scavenging harmful free radicals [68].

Hyperglycemia, hyperinsulinemia, and hyperlipidemia coexist in patients with T2D. Indeed, hyperglycemia can promote lipogenesis at least in the liver. O-GlcNAcylation, an important glucose-dependent posttranslational modification, stabilizes carbohydrate responsive element binding protein (ChREBP) and increases its transcriptional activity, thus promoting lipogenesis, through upregulating lipogenic genes such as acetyl-CoA carboxylase and fatty acid synthase. OGT (O-GlcNAc transferase) overexpression increased ChREBP in mouse liver, leading to fatty liver. However, OGA (O-GlcNAcase) overexpression also reduced ChREBP and therefore decreased lipogenesis, and improved lipid profile of OGA-treated db/db mice [73].

### 2.7. Chronic Inflammation

Chronic low-grade inflammation has been observed in obesity, T2D, CVD, and other MetS-related chronic diseases. It is widely established that immune cells play an important role in this pathogenesis. Metabolic disturbances activate the immune system and result in immune cells activation in tissues such as the adipose, liver, pancreas, and vasculature. Systematically it increases plasma inflammatory markers, such as TNFα, IL-6, IL-1b, etc. [74]. Among the immune cells, macrophages polarized activation has drawn much attention in the last decades and seems to play a crucial role in local and systemic chronic inflammation [75,76,77].

Adipose tissue macrophages have been studied increasingly in recent years and have been shown to be a key contributor to adipose inflammation and systemic inflammation. Adipose is not only for fat storage, but also a powerful autocrine and endocrine organ. Under some conditions, such as fat accumulation, fat cells secrete not only adipokines but also cytokines, such as TNFα and MCP1, etc. These signals attract monocytes in circulation and recruit them to local adipose tissue. Here the monocytes differentiate to macrophages, infiltrating adipose tissue, particularly the surrounding fat cells, forming a crown-like structure [78]. Different macrophage subpopulations may exhibit a scale of different properties such as a two polar of function: pro-inflammatory or anti-inflammatory. Upon classically pro-inflammatory activation (called M1), usually by LPS, TNFα, macrophages can produce more pro-inflammatory cytokines to exacerbate inflammation. When macrophages are alternatively activated (called M2), usually by IL-4, they produce anti-inflammatory cytokines such as IL-10 which ameliorate inflammation and assists in tissue repair. TNFα decreases insulin sensitivity. Nuclear factor kappa-light-chain-enhancer of activated B cells (NFκB) and JNK phosphorylation are the main pathways involved in inflammatory responses, so they are widely used as indicators of inflammation. Evidence has demonstrated that local adipose inflammatory responses contribute to local insulin resistance, and further contribute to systemic inflammation and insulin resistance. Many studies have been conducted in an effort to find the modulator/regulator of macrophage activation in attempts to control the macrophage activation pattern. It is believed this is either by M1 or M2. By switching M1 to M2, inflammation can be reversed and insulin resistance can be ameliorated [78]. T cells have also been shown to play a similar role in inflammation.

We previously found that heme oxygenase-1 (Hmox1 or HO-1) is a target of FOXO1 in the liver impairing mitochondrial biogenesis and function [79]. Hmox-1 is highly expressed in the cell in response to oxidative stress. It is an enzyme that catalyzes the degradation of heme that produces biliverdin, ferrous iron, and carbon monoxide. Heme is an essential component for mitochondrial electron transport chain. There is evidence that levels of heme oxygenase are positive predictors of metabolic disease, insulin resistance, and metaflammation. This is supported by a recent study demonstrating that HO-1 is one of the strongest positive predictors of metabolic disease in both mice and humans. Conditional HO-1 deletion in mice, either hepatocytes or macrophages, protects mice from HFD-induced inflammation and insulin resistance. The reduced meta-inflammation upon HO-1 deletion dramatically reduced metabolic disease, such as steatosis [80].

Research in recent decades has uncovered the pivotal role of toll-like receptors (TLRs), especially TLR2 and TLR4, in chronic inflammation, insulin resistance, and pathogenesis of obesity and MetS. TLRs can serve as effective therapeutic targets to reverse diabetes and MetS [81]. The inflammatory signaling cascades initiate activation of NFκB, JNK, and inflammasomes, and interfere with insulin signaling. NFκB signaling in different tissues, such as adipose tissue, liver, hypothalamus, skeletal muscle, endothelial cells, and macrophages, contributes to the development of obesity and related MetS [82].

Suppression of obesity-associated inflammation in different tissues, including adipose tissue, liver, intestine etc. by different nutritional interventions can operate separately or synergistically to ameliorate systemic insulin sensitivity and metabolic homeostasis [83]. Omega-3 (ω-3) fatty acids have been proposed to serve as a dietary intervention for reducing obesity-associated inflammation and insulin resistance [84,85]. However, some research results indicated that the anti-inflammatory benefits of ω-3 fatty acids are not necessarily associated with a decrease in body weight or improvement of insulin sensitivity [86,87]. As controversial results coexist, more research needs to be done to fully investigate the benefits of ω-3 fatty acids in MetS prevention and intervention. A research group has recently discovered ω-3 PUFA’s exciting potential to block the auto-immunity, restore β-cell regeneration, and sharply reduce the incidence of T1D [88]. Whether the anti-inflammatory effect of ω-3 PUFA can achieve such an improvement in T2D, remains to be investigated.

### 2.8. Circadian

Our body is under a 24-hour cycle which controls the rhythm of many physiological processes. This clock is intrinsic and influenced by external cues, such as the sun, temperature etc. The driver (core loop) consists of the positive elements circadian locomotor output cycles kaput (CLOCK) and aryl hydrocarbon receptor nuclear translocator-like protein 1 (BMAL1), and negative feedback elements period (PER) and cryptochrome (CRY). Therefore, it is also important to understand when the metabolic processes happen other than “what” and “how” they happen. For example, individuals with insomnia tend to be more obese. Obese mice displayed disrupted circadian. Vice versa, circadian disrupted mice gained more weight when fed with HFD. When circadian was disrupted, macrophage inflammatory responses exacerbated resulting in more severe insulin resistance in mice fed with HFD [78]. The concept of chronobiological-based therapies was brought up to reset the circadian rhythm among obese individuals [89].

The CLOCK transcriptional factor is a vital component of circadian clock. The homozygous *clock* mutant mice developed a metabolic syndrome of hyperglycemia hyperlipidemia and overweight, suggesting circadian plays an important role in energy balance [90]. A recent study demonstrated that insulin-Forkhead box class O3 (FOXO3) signaling pathway is required for circadian in the liver through regulation of *clock*, indicating *clock* as a downstream target of FOXO3 [91]. Another essential component of circadian clock is BMAL1, which is also involved in glucose homeostasis [92]. With knock down or disruption of BMAL1, gluconeogenesis was severely abolished and insulin resistance occurred [92,93]. Disruption of circadian clock alters the metabolic homeostasis, which can result in metabolic syndrome [94,95,96].

On the other hand, circadian clock can be reprogrammed by nutritional challenge and diseases. High fat diet caused the impaired CLOCK:BMAL1 chromatin recruitment and altered the clock synchronization to light [97,98,99]. In streptozotocin (STZ)-induced diabetic rats, the clock in heart lost normal synchronization with the environment [100]. Taken together circadian clock and metabolic syndrome are closely linked. In the future, new therapeutic methods for obesity and type 2 diabetes should take circadian clock into a consideration.

### 2.9. Genetics and Epigenetics

Gene structure and function can be influenced by the environment. In the Greenland Inuit population, fatty acid desaturases (FADS1, 2, 3) are suspected to have been selection-driven by a diet high in polyunsaturated fatty acides (PUFAs) during their environmental diet adaptation [101]. It is known that genetic factors, interacting with the environment, contribute to MetS. Detecting these specific genes associated with the disease or modulating genes related to the environment can be two strategies for gene therapy. As techniques of gene sequencing and editing keep developing, the cost becomes more affordable, allowing more research and application in gene diagnosis, edit, and therapy. In the meantime, gene-nutrition interaction (nutrigenomics) has attracted more attention and has innovated the field of personalized nutrition. Women with the genotype of IRS1-rs2943641 TT exhibit reduction of insulin resistance and T2D risk when circulating vitamin D-25(OH)D is higher. The beneficial effect of high circulating 25(OH)D for carriers of the major allele (rs2943641 C) is not as strong. Differential Vitamin D supplementation levels have the potential to be applied to people based on their genotype, however more research is needed to confirm this theory [102].

As previously mentioned, IRS protein tyrosine and threonine/serein phosphorylation can determine insulin sensitivity. Recently, research has shown that HFD can enhance acetylation of a number of proteins, of which one is p300. This is a global transcriptional cofactor that enhances FOXO1-mediated gene expression [103], acetylates IRS1, 2, and subsequently impairs IRS interaction with insulin receptors, resulting in insulin resistance [104]. These results tell us that diets and nutrients can modify proteins and regulate their functionality in control of metabolism in the cells and body.

### 2.10. Gut Microbiota

Gut microbial imbalance has been observed in obese people. According to Remley et al. alterations in gut microbiota affect various epigenetic patterns of gene expression involved in metabolic and inflammatory homeostasis [105]. HFD disrupts the structure of gut microbiota and causes inflammation—an important contributor to HFD-induced MetS [106]. To investigate which exact factor—fat content or other nutrients—in HFD drives adiposity compared to normal chow diet (NCD), Benoit et al. compared 14 compositionally defined diets (CDD) with different fat content, protein sources, and fiber source combinations. It has been suggested that HFD-induced obesity is greatly promoted by its lack of soluble fiber (inulin). Inulin is an important ingredient that supports microbiota-mediated intestinal tissue homeostasis, preventing inflammation and MetS [107]. A recent pig study also proved that feeding inulin significantly limits the effects of HFD on the microbiota, resulting in more diverse microbial populations, increased fatty acid oxidation, and suppressed fatty acid synthesis [108].

Probiotic supplementation seems to be effective to improve and even prevent diet-induced MetS phenotype. Three probiotic strain supplementations in HFD-fed mice all attenuated MetS, and shifted the overall structure of the HFD-disrupted gut microbiota toward that of lean mice on normal chow diet [109]. In a human study with a small sample size, probiotic supplementation for four weeks with the last week shifting diet to high-fat, high-energy (50% increase in energy intake) can help prevent high-fat and overfeeding induced insulin resistance, compared to a group with no probiotic supplementation [110]. Moreover, probiotic supplementation benefits hypertension through improvement of lipid profiles, regulation of insulin sensitivity, and bioconversion of bioactive isoflavones [111]. More double-blinded randomized trials with larger sample sizes are warranted, as well as an examination of the different species of probiotics used.

### 2.11. Dietary Effects

Nutrition is a key environmental factor for metabolic syndrome. A study in healthy men suggested that the initial event caused by overnutrition may be oxidative stress but not inflammatory or ER stress, which in part, promotes carbonylation and inactivation of GLUT4, resulting in insulin resistance [112]. To limit calorie intake, some recommended avoiding high-fat or regular cheese but to take the reduced-fat substitute. However, substitution of high-fat cheese by reduced-fat cheese will not improve LDL cholesterol or MetS risk factors [113,114,115]. Changing only one part of the eating pattern without controlling the rest will not have significant health results. Researchers recently designed a study using a 4-day fasting mimicking diet (FMD) which constituted a low-calorie, low-protein, low-carbohydrate but high-fat diet. The goal was to cause changes similar to those caused by water-only fasting, while considering the challenges and side effects associated with prolonged fasting. It is suggested that FMD cycles promote β-cell regeneration, restore insulin secretion and glucose homeostasis in both type 1 and 2 diabetic mouse models, as well as in T1D patients. The underlying mechanism is likely related to the fact that fasting conditions reduce PKA and mTOR activity, induce Sox2 and Ngn3 expression, and increase insulin production in pancreatic islets [116].

Saturated or unsaturated fats can have different outcomes on insulin resistance and CVD complications. Frank Hu’s team found that replacing 5% of energy intake from saturated fatty acid with equivalent energy intake from PUFA, monounsaturated fatty acids (MUFA), or high-quality carbohydrate (whole grains, but not refined starches or added sugars) can lower risk of CVD by 25%, 15%, and 9%, respectively [117]. Additionally, it has been found that polyunsaturated fats have cardio-protective benefits from CVD and hypertension in humans [118].

The exact mechanisms underlining these beneficial effects of unsaturated fats are not completely understood yet, but most likely they are related to their interference with inflammation. A recent study suggested that dietary MUFA can impede adipose NLRP3 inflammasome-mediated IL-1β secretion, and attenuate insulin resistance. Additionally adipose dysfunction is disrupted via the preservation of AMP-activated protein kinase (AMPK) activity, even though these mice maintained diet-induced obesity [119]. Branched chain amino acids (BCAAs) have also been reported as being beneficial for the improvement of obesity and T2D. However, increased levels of BCAAs in circulation can also serve as a marker for lack of insulin activity due to metabolic dysfunction. Thus BCAA levels do not serve as reliable sources of evaluation [120]. The specific type of fibers may be needed to be clarified in the future due to the protective effect found in consuming soluble fiber (inulin) in HFD. For example, this effect is not seen when soluble fiber is replaced with insoluble fiber (cellulose) [107].

With the extended utilization of dietary supplements, people realize benefits for metabolic disease, such as the polyphenols. Polyphenols can improve the disrupted glucose homeostasis in the insulin resistance state and exert hypolipidemic effects [121,122,123,124,125,126]. However, overdose effects can be a very serious issue. These have to be evaluated extensively, in order to give the public more specific guidance. As previously discussed, this over-macro-nutrition has many detrimental effects for cells and health. In addition, micronutrients also control multiple metabolic processes and enzymatic activities, which is not discussed in this review. For example, selenium (Se), is an essential trace element, and is recommended worldwide for supplementation to prevent Se-deficient pathological conditions, including diabetes and insulin resistance. However, it is indicated that overdose of this micronutrient also increases ROS and impairs hepatic insulin sensitivity [127].

## 3. Conclusions

MetS has a profound impact on the development of diabetes and CVD. Obesity can play a major role in the development of other features in MetS. IRS proteins and associated signaling cascades can play central roles in the control of nutrient metabolism and organ function. Restriction in food intake and quality selection of nutritional components have huge impacts on insulin secretion and responses, the cellular redox and inflammatory states, and bodily insulin sensitivity. Managing an appropriate balance of energy requirement of cells and bodily inflammation will be crucial for insulin sensitivity and control of MetS and disease development in the future.

## Figures and Tables

**Figure 1 ijms-20-00128-f001:**
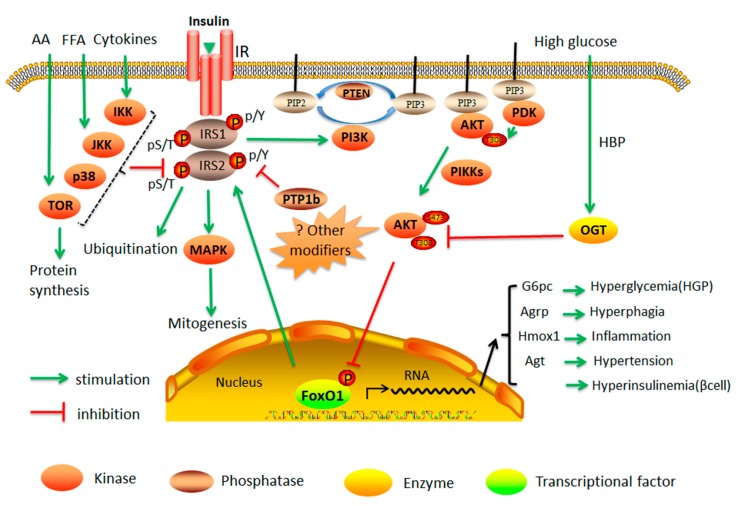
The role of kinases in the insulin signaling cascades and its interaction with nutrients in control of FOXO1-mediated physiological function. AA—amino acids; FFA—free fatty acids; HGP—hepatic glucose production; Agt—angiotensinogen; Hmox-1—heme oxygenase-1; G6pc—glucose-6-phosphatase catalytic subunit; Agrp—agouti-related peptide; pY—tyrosine phosphorylation; pS/T—serine/threonine phosphorylation; OGT—O-GlcNAc transferase; HBP—hexosamine biosynthetic pathway; PIP2—phosphatidylinositol-4,5-biphosphate; PIP3—phosphatidylinositol-3,4,5-triphosphate; PTP1B—protein tyrosine phosphatase; PTEN—phosphatase and tensin homolog; PDK—phosphoinositide-dependent protein kinase; PI3K—phosphatidylinositide 3-kinase; PIKKs—PI3K-related kinase family; IR—insulin receptor.

**Table 1 ijms-20-00128-t001:** Development of clinical criteria of metabolic syndrome by different health organizations.

Evaluation Content	WHO 1998	EGIR 19 99	NCEPATPIII 2001	AACE 2003	NCEP ATPIII 2005	AHA/NHLBI 2005	IDF 2005	IDF/NHLBI 2009
Criteria	IGT, IFG, T2D, or reduced insulin sensitivity plus any two of the following	Plasma insulin > 75 percentile plus any two of the following	Any three of the following	IGT or IFG plus any of the following	Any three of the following	Any three of the following	Increased WC plus any two of the following	Three out of five of the following
Obesity	Men: WHR > 0.9; Women: WHR > 0.85 and/or BMI > 30 kg/m^2^	WC ≥ 94 cm in men or ≥ 80 cm in women	WC ≥ 102 cm in men or ≥ 88 cm in women	BMI > 25 kg/m^2^	WC ≥ 102 cm in men or ≥ 88 cm in women	WC ≥ 102 cm in men or ≥ 88 cm in women	population-specific increased WC cutoffs	population-and country-specific WC cutoffs
Glucose	IGT, IGF, or T2D	IGT or IFG	≥110 mg/dL (including T2D)	IGT or IFG	≥100 mg/dL (including T2D)	≥100 mg/dL or on drug treatment for elevated glucose	≥100 mg/dL (including T2D)	≥100 mg/dL
Triglycerides (TG)	TG ≥ 150 mg/dL	TG ≥ 150 mg/dL	TG ≥ 150 mg/dL	TG ≥ 150 mg/dL	TG ≥ 150 mg/dL or on therapy lowering TG	TG ≥ 150 mg/dL or on drug treatment for elevated triglycerides	TG ≥ 150 mg/dL or on therapy lowering TG	TG ≥ 150 mg/dL
High density lipoprotein (HDL)-cholesterol (HDL-C)	HDL-C < 40 mg/dLin men or HDL-C < 50 mg/dL in women	HDL-C < 39 mg/dL in men or women	HDL-C < 40 mg/dL in men or HDL-C < 50 mg/dL in women	HDL-C < 40 mg/dL in men or HDL-C < 50 mg/dL in women	HDL-C <40 mg/dL in men or HDL-C < 50 mg/dL in women on therapy increasing HDL-C	HDL-C < 40 mg/dL in men or HDL-C < 50 mg/dL in women or on drug treatment for reduced HDL-C	HDL-C < 40 mg/dL in men or HDL-C < 50 mg/dL in women on therapy increasing HDL-C	HDL-C< 40 mg/dL in men or HDL-C < 50 mg/dL in women
Blood pressure	≥140/90 mmHg	≥140/90 mmHg or on antihypertensive therapy	≥130/85 mmHg	≥130/85 mmHg	≥130/85 mmHg or on antihypertensive therapy	≥130/85 mmHg or on antihypertensive therapy	≥130/85 mmHg or on antihypertensive therapy	≥130/85 mmHg or on antihypertensive therapy

IGT, impaired glucose tolerance, IFG, impaired fasting glucose, TG, triglycerides, T2D, type 2 diabetes, WC, waist circumference, WHR, waist/jip ratio. WHO, World Health Organization. EGIR, European Group for the study of Insulin Resistance rename “insulin resistance syndrome”. NCEP ATPIII, National Cholesterol Education Program, Adult Treatment Panel III, “Metabolic Syndrome” reassigned. AACE, American Association of Clinical Endocrinologists, “Insulin Resistance Syndrome”. IDF, International Diabetes Federation. AHA/NHLBI, American Heart Association/National Heart, Lung and Blood Institute.

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
