# Peer review of "Etiology of Metabolic Syndrome and Dietary Intervention"

_ijms, 2018, doi:10.3390/ijms20010128_

Reviewer 1 Report

The manuscript by Xu et al. is a narrative review about mechanisms of metabolic syndrome (MetS) pathogenesis. Overall, the scope of this review is too broad for the limited space, and therefore description in each section are too brief. I don't see the point of publishing this article in the current form. A potentially interesting point is, however, IRS modification and its downstream signaling pathways as the authors also stated in Conclusion, Figure, and title. I would recommend the authors to focus on this aspect, which in my opinion makes this manuscript more unique and up-to-date.

This manuscript has many grammatical mistakes and incomplete sentences. I suggest authors to have a professional English editing before revision. Also, spaces are missing in many parts of the manuscript.

Please use a consistent abbreviation rule throughout the paper. For example, IRS and PTP1B are spelled out, but PI3K, PDK etc. are not. BCF is spelled out twice; PIP3 is spelled out in section 2-4 while it appears in section 2-3. FOXO1 in section 2-3, but Foxo1 in 2-5.

Specific comments

Introduction

1) Last paragraph: Imcomplete sentence "As metabolic syndrome is a crucial contributor for type 2..."

Section 2-1.

2) This section well describes insulin action and insulin resistance, but misses the relationship between MetS and insulin resistance. Which state of insulin resistance is commonly observed in MetS? Also, more references are required in this section including recent evidence. Currently this section looks like a textbook description.

Section 2-3.

3) Does Akt bind to PIP3? The reviewer does not know their direct binding, and it is not shown in Fig. 1 either. Please edit the sentence or provide reference(s).

4) GLUT4 actually requires the activation step after translocation, and this step is also inhibited under insulin resistance.

5) Page 4, line 1: "involved in"

Section 2-4.

6) This section mainly focuses on the function of IRS in insulin signaling along with binding proteins and Ser/Thr phosphorylation. I think that the authors miss several important publications, which is summarized in Hakuno et al. Front Endocrinol 6, Article 73, 2015.

7) autophage -> autophagy?

Section 2-5.

8) The last sentence of the first paragraph should be revised.

9) Brain is not the only target of leptin. 

Section 2-8.

10) Circadian certainly plays important roles in MetS, but again the description in this section is too brief. As stated above, it would be better to focus of some specific molecule, such as IRSs, and make a section only if the topic is relavant to the molecule.

Table 1. I guess that the authors included only important definitions of the metabolic syndrome since for some definitions (such as Reaven 1988 and EGIR 1999) all cells remain blank. I understand the importance of definitions detailed here, but still feel it is strange as a Table. I would recommend the authors to use footnote to describe Reaven, EGIR, NCEP ATPIII, and AHA/NHLBI.

Author Response

Reviewer 1:

The manuscript by Xu et al. is a narrative review about mechanisms of metabolic syndrome (MetS) pathogenesis. Overall, the scope of this review is too broad for the limited space, and therefore description in each section are too brief. I don't see the point of publishing this article in the current form. A potentially interesting point is, however, IRS modification and its downstream signaling pathways as the authors also stated in Conclusion, Figure, and title. I would recommend the authors to focus on this aspect, which in my opinion makes this manuscript more unique and up-to-date.

This manuscript has many grammatical mistakes and incomplete sentences. I suggest authors to have a professional English editing before revision. Also, spaces are missing in many parts of the manuscript.

Please use a consistent abbreviation rule throughout the paper. For example, IRS and PTP1B are spelled out, but PI3K, PDK etc. are not. BCF is spelled out twice; PIP3 is spelled out in section 2-4 while it appears in section 2-3. FOXO1 in section 2-3, but Foxo1 in 2-5.

Specific comments

Introduction

1) Last paragraph: Imcomplete sentence "As metabolic syndrome is a crucial contributor for type 2..."

Response: We revised this paragraph. “Metabolic syndrome is a crucial contributor for type 2 diabetes and cardiovascular disease.”

Section 2-1.

2) This section well describes insulin action and insulin resistance, but misses the relationship between MetS and insulin resistance. Which state of insulin resistance is commonly observed in MetS? Also, more references are required in this section including recent evidence. Currently this section looks like a textbook description.

Response: Thanks for this comment and this is a very important question. We revised this section and added the relationship between MetS and insulin resistance. MetS is also widely known as insulin resistance syndrome due to the causative role insulin resistance plays in the syndrome. Even the group GEIR and AACE used the term insulin resistance syndrome rather than a metabolic syndrome. Because insulin resistance is difficult to evaluate directly, especially in a clinical setting, some types of evidence were accepted, including impaired fasting glucose (IFG), impaired glucose tolerance (IGT) and type 2 diabetes mellitus. Some other factors were also served as diagnostic criteria, such as blood pressure, obesity, and disordered lipid profile (Table 1).

Section 2-3.

3) Does Akt bind to PIP3? The reviewer does not know their direct binding, and it is not shown in Fig. 1 either. Please edit the sentence or provide reference(s).

Response: Yes, Akt can directly bind to PIP3. We revised the Figure and added related references in this section.

4) GLUT4 actually requires the activation step after translocation, and this step is also inhibited under insulin resistance.

Response: Thank you very much for your comments and this question. We revised this section and some references were added. GLUT4 activation by insulin is also important during the glucose disposal, and p38 MAPK may be involved in this process. Under insulin resistance state, both GLUT4 translocation and activation are affected.

5) Page 4, line 1: "involved in"

Response: Thanks for the patience of the reviewer. We revised this sentence.

Section 2-4.

6) This section mainly focuses on the function of IRS in insulin signaling along with binding proteins and Ser/Thr phosphorylation. I think that the authors miss several important publications, which is summarized in Hakuno et al. Front Endocrinol 6, Article 73, 2015.

Response: This is a very important point and thank you for pointing out this. We revised and added some references related to IRS associated protein. The high-molecular-mass complexes containing insulin receptor substrates also involved in mediating and regulating insulin-like activities are summarized in a previous review. Not only insulin-like growth factors (IGF)/insulin but also other cytokines/hormones contribute to the formation of IRSs associated with other proteins (IRSAPs). IRSAPs can regulate IGF/insulin signaling pathway by controlling IRS tyrosine phosphorylation and interaction with PI3K.  IRSAPs also play important roles in the modification of IRSs stability, intracellular localization, and RNA metabolism and translation.    

7) autophage -> autophagy?

Response: Yes. We revised this mistake. Thanks for your correction.

Section 2-5.

8) The last sentence of the first paragraph should be revised.

Response: Thanks for your question. We modified the last sentence. Saturated fatty acids, such as palmitate, reduces IRS1, 2 tyrosine phosphorylation, induces serine/threonine phosphorylation by activation of intracellular protein kinases, such as PKC and JNK, and promotes FOXO1 activity.

9) Brain is not the only target of leptin.

Response: We totally agree with the reviewers and some references were added. Leptin may also regulate glucose homeostasis, pancreases β cells, and insulin-sensitive tissues.

Section 2-8.

10) Circadian certainly plays important roles in MetS, but again the description in this section is too brief. As stated above, it would be better to focus of some specific molecule, such as IRSs, and make a section only if the topic is relavant to the molecule.

Response: Yes, this section is too brief. However, we address the etiology of MetS, in

this short review and circadian plays important role in this process. We add two

paragraphs to explain the interaction of circadian clock and metabolic syndrome.

But your suggestion is very meaningful and constructive. 

Table 1. I guess that the authors included only important definitions of the metabolic syndrome since for some definitions (such as Reaven 1988 and EGIR 1999) all cells remain blank. I understand the importance of definitions detailed here, but still feel it is strange as a Table. I would recommend the authors to use footnote to describe Reaven, EGIR, NCEP ATPIII, and AHA/NHLBI.

Response: Thanks for the advice of the reviewer. And we added the information and use the footnote to describe NCEP ATP III etc.

Reviewer 2 Report

The manuscript is a review of the various demonstrated and proposed causes of metabolic syndrome (MetS). While the manuscript is well-written, the following changes are suggested:

1) Title: The title is misleading as it gives an impression that the manuscript will solely focus on kinase signaling. The authors should rewrite the title to better reflect the content of the manuscript.

2) The manuscript is full of grammatical errors: incorrect word usage, incomplete sentences, missing spaces between words. This makes it very difficult to read the manuscript. These mistakes must be corrected before the manuscript is published.

Author Response

Reviewer 2:

The manuscript is a review of the various demonstrated and proposed causes of metabolic syndrome (MetS). While the manuscript is well-written, the following changes are suggested:

1) Title: The title is misleading as it gives an impression that the manuscript will solely focus on kinase signaling. The authors should rewrite the title to better reflect the content of the manuscript.

Response: Thanks, and we renamed our manuscript. Etiology of Metabolic Syndrome and Dietary Intervention.

2) The manuscript is full of grammatical errors: incorrect word usage, incomplete sentences, missing spaces between words. This makes it very difficult to read the manuscript. These mistakes must be corrected before the manuscript is published.

Response: Thanks for the comments. We have revised the manuscript and errors were amended. We marked the changes in the manuscript. 

Round  2

Reviewer 1 Report

I read the revised manuscript and consider that the authors significantly improved the manuscript. Now I would recommend this paper for publication.